# High-Risk Neutropenic Fever and Invasive Fungal Diseases in Patients with Hematological Malignancies

**DOI:** 10.3390/microorganisms12010117

**Published:** 2024-01-06

**Authors:** Giovanni Mori, Sara Diotallevi, Francesca Farina, Riccardo Lolatto, Laura Galli, Matteo Chiurlo, Andrea Acerbis, Elisabetta Xue, Daniela Clerici, Sara Mastaglio, Maria Teresa Lupo Stanghellini, Marco Ripa, Consuelo Corti, Jacopo Peccatori, Massimo Puoti, Massimo Bernardi, Antonella Castagna, Fabio Ciceri, Raffaella Greco, Chiara Oltolini

**Affiliations:** 1Infectious Diseases Unit, Vita-Salute San Raffaele University, 20132 Milan, Italy; giovanni.mori@apss.tn.it (G.M.);; 2Infectious Diseases Unit, Ospedale Santa Chiara, 38122 Trento, Italy; 3Infectious Diseases Unit, IRCCS San Raffaele Scientific Institute, 20127 Milan, Italy; 4Haematology and Bone Marrow Transplant Unit, IRCCS San Raffaele Scientific Institute, 20132 Milan, Italy; 5Centre for Immuno-Oncology, National Cancer Institute, Eliminate NIH, Bethesda, MD 20850, USA; 6Infectious Diseases Unit, ASST Grande Ospedale Metropolitano Niguarda, 20161 Milan, Italy; 7Faculty of Medicine and Surgery, University of Milano-Bicocca, 20126 Milan, Italy

**Keywords:** neutropenic fever, invasive fungal diseases, acute leukaemia, allogeneic hematopoietic stem cell transplantation, antifungal prophylaxis

## Abstract

Invasive fungal diseases (IFDs) still represent a relevant cause of mortality in patients affected by hematological malignancies, especially acute myeloid leukaemia (AML) and myelodysplastic syndrome (MDS) undergoing remission induction chemotherapy, and in allogeneic hematopoietic stem cell transplantation (allo-HSCT) recipients. Mold-active antifungal prophylaxis (MAP) has been established as a standard of care. However, breakthrough IFDs (b-IFDs) have emerged as a significant issue, particularly invasive aspergillosis and non-*Aspergillus* invasive mold diseases. Here, we perform a narrative review, discussing the major advances of the last decade on prophylaxis, the diagnosis of and the treatment of IFDs in patients with high-risk neutropenic fever undergoing remission induction chemotherapy for AML/MDS and allo-HSCT. Then, we present our single-center retrospective experience on b-IFDs in 184 AML/MDS patients undergoing high-dose chemotherapy while receiving posaconazole (*n* = 153 induction treatments, *n* = 126 consolidation treatments, *n* = 60 salvage treatments). Six cases of probable/proven b-IFDs were recorded in six patients, with an overall incidence rate of 1.7% (6/339), which is in line with the literature focused on MAP with azoles. The incidence rates (IRs) of b-IFDs (95% confidence interval (95% CI), per 100 person years follow-up (PYFU)) were 5.04 (0.47, 14.45) in induction (*n* = 2), 3.25 (0.0013, 12.76) in consolidation (*n* = 1) and 18.38 (3.46, 45.06) in salvage chemotherapy (*n* = 3). Finally, we highlight the current challenges in the field of b-IFDs; these include the improvement of diagnoses, the expanding treatment landscape of AML with molecular targeted drugs (and related drug–drug interactions with azoles), evolving transplantation techniques (and their related impacts on IFDs’ risk stratification), and new antifungals and their features (rezafungin and olorofim).

## 1. Introduction

Major improvements have been achieved in the prophylaxis, treatment and outcome of invasive fungal diseases (IFDs); however, they still represent an important cause of mortality in patients affected by hematological malignancies (HM) who are undergoing remission induction chemotherapy for acute leukaemia and allogeneic hematopoietic stem cell transplantation (allo-HSCT). Patients with long-lasting neutropenia, such as patients with acute myeloid leukaemia (AML) or myelodysplastic syndrome (MDS), continue to represent the population at highest risk of developing neutropenic fever (NF) and IFDs [1]. The most frequently identified fungal pathogens are *Aspergillus* spp. and *Candida* spp., which significantly contribute to mortality in these patients [2,3,4,5,6]; therefore, mold-active antifungal prophylaxis (MAP) has been established as a standard of care [7,8,9,10,11,12]. However, breakthrough IFDs (b-IFDs) in patients receiving antifungal prophylaxis (AFP), including MAP, is an issue for clinicians, alongside with the rising resistance to antifungals and the continuous change in IFD epidemiology, due to both the increased use of mold-active drugs and the improved survival of HM patients.

## 2. Search Strategy

In this narrative review, we discuss the literature of the last decade that concerns IFDs in HM patients with NF who are undergoing induction chemotherapy for acute leukaemia and allo-HSCT.

A structured literature search in PubMed/MEDLINE database was performed in October 2023 (update freezing date: 31 October) of all English-written full articles with the following search terms: (“Hematopoietic Stem Cell Transplantation” OR “Acute Leukaemia”) AND (“Invasive Fungal Infections” OR “Breakthrough Invasive Fungal Infections” OR “Antifungal Prophylaxis”). Studies published prior to 2010 (except randomized controlled trials; RCTs), including pre-prints, abstracts, letters, reviews, commentary articles, opinion articles and animal studies were excluded. This timeframe was selected to include more recent studies that highlight an updated epidemiology and outcome of IFDs. Fifty studies were included, all assessed to be of moderate to high quality and with moderate risk of bias.

Then, we report the single-center experience of San Raffaele Scientific Institute on b-IFDs in AML/MDS patients undergoing induction, consolidation and salvage chemotherapy while receiving posaconazole.

## 3. Diagnosis and Definitions of Invasive Fungal Diseases

The diagnosis of IFDs, especially the mycological one, is a crucial issue for the management of patients with high-risk NF. A recent trial highlights the pursuit of diagnosis, demonstrating the safety of a pre-emptive strategy compared to empiric antifungal therapy (AFT), in 549 AML/MDS patients undergoing induction chemotherapy or allo-HSCT with fluconazole prophylaxis, who are randomly assigned to receive caspofungin empirically or pre-emptively. The overall survival (OS) at day 42 was non-inferior (pre-emptive arm 96.7%, empiric arm 93.1%) and IFD rates at day 84 were superimposable (pre-emptive arm 7.7%, empiric arm 6.6%), while the exposure to caspofungin was significantly lower in the pre-emptive arm (27% vs. 63%), without excess mortality [13].

In 2020, the European Organization for Research and Treatment of Cancer and the Mycoses Study Group Education (EORTC/MSG) consensus group updated the IFD definitions published in 2008 [14], and we highlight the main revisions [15].

### 3.1. Proven Invasive Fungal Diseases

Proven IFDs apply to patients regardless of their immune status. The culture-based and histopathological detection of fungi, in a specimen from a normally sterile site, remains the gold standard for diagnosis. Wherever feasible, when fungi are seen in formalin-fixed paraffin-embedded tissue, the amplification of fungal DNA by a polymerase chain reaction (PCR) combined with DNA sequencing has become a criterion for proven diagnosis.

### 3.2. Probable Invasive Fungal Diseases

Probable IFDs require a host risk factor, alongside clinical, radiological and mycological findings.

The main revisions of their radiological features are as follows:Among patients with pulmonary invasive aspergillosis (IA), nodules with halo sign remain useful among neutropenic patients, but they are nonspecific for IA in other groups. Conversely, in invasive mucormycosis (IM), consolidation is the most frequent presentation, followed by mass lesions and nodules. Multiple nodules (>10) and pleural effusions appear more frequently in IM than in IA and reverse halo sign is more specific for IM.Among non-neutropenic patients, multiple nodules and various nonspecific findings are present (bronchopneumonia, consolidation, cavitation, pleural effusions, ground glass or tree-in-bud opacities and atelectasis).

Below are the main revisions of the host factors, including new additions:Hematologic malignancy (acute leukaemia, lymphoma, multiple myeloma) intended as active cancer receiving treatment or remission in the recent past.Receipt of a solid organ transplant (SOT).Treatment with B-cell immune-suppressants, such as Bruton’s tyrosine kinase inhibitors.Inherited severe immunodeficiency, not only chronic granulomatous disease, but also STAT-3 or CARD9 deficiency, STAT-1 gain of function or severe combined immunodeficiency.Steroid-refractory grade III–IV acute graft-versus-host disease (a-GVHD) involving the gut, lungs or liver.

The main revisions of the mycological criteria, according to the new available literature:*Aspergillus* galactomannan (GM) detection using serum, bronco-alveolar lavage (BAL) and cerebrospinal fluid (CSF) remains essential. The thresholds of the Platelia *Aspergillus* assay have been standardized to support the diagnosis of IA as follows: single serum ≥ 1, BAL ≥ 1, CSF ≥ 1 or single serum ≥ 0.7 plus BAL ≥ 0.8. The performance of GM remains clearly lower in non-neutropenic patients and those receiving MAP.*Aspergillus* PCR assays obtain sufficiently robust data to be performed on plasma, serum, whole blood and BAL from adults with HM or from allo-HSCT recipients for screening and diagnosis confirmation. Any of the following is a criterion for probable IA: ≥2 consecutive positive tests on plasma, serum or whole blood; ≥2 positive tests on BAL or ≥1 positive test on plasma, serum or whole blood plus one positive test on BAL. There are few commercial tests (mostly in-house assays); despite technologic variability, PCR performance is comparable with that for detecting GM. The Fungal PCR Initiative developed recommended criteria for *Aspergillus* PCR test rather than a standardized method per seBeta-D-glucan (BDG) detection is suitable for diagnosing IFDs in specific settings (HM with and without neutropenia, pre-engraftment phase after allo-HSCT, intensive care unit patients at high risk for invasive candidiasis (IC)). The recommended assay is Fungitell, with a single threshold (>80 pg/mL). Confidence for true positive results increases with repeated positive tests (≥2 consecutive positive serum samples constitute a criterion for probable IC) and for values that greatly exceed the threshold. BDG displays an extremely high negative predictive value (NPV) for candidiasis, aspergillosis, pneumocystosis or fusariosisT2Candida, performed on blood specimens, has a very high NPV and a variable positive predictive value (PPV) (62% in patients with sepsis, shock or an intensive care stay of >3–7 days; 92% in neutropenic leukemic patients or post allo-HSCT while not receiving AFP). It represents a criterion for diagnosis of probable IC.

## 4. Advances in Microbiological Diagnostic Tools and Radiological Techniques

After the update to the EORTC/MSG criteria [15], further studies have been published on new diagnostic tools, to expand our ability to diagnose IFDs in high-risk NF patients, especially mycological tests (PCR-based molecular methods, plasma microbial cell-free DNA sequencing (mcfDNA-Seq)) and radiological ones ([¹⁸F]FDG-PET-CT scan).

Starting from molecular tests, the *AsperGenius* PCR assay, which detects *A. fumigatus*, *A. terreus*, other *Aspergillus* species and mutations in the *A. fumigatus* cyp51A gene (TR34/L98H, TR46/T289A/Y121F), was evaluated in a prospective study on 323 HM patients with NF and pulmonary infiltrates, who were undergoing BAL within 48 h, of which 36% were diagnosed as having probable IA. The assay turned out positive in 40% of BAL, more frequently in samples with increasing GM positive values, and *A. fumigatus* DNA was detected in 30%. PCR, used to detect mutations associated with azole-resistant *A. fumigatus*, was conclusive in 58/89 samples and positive in 8/58 (1/8 treatment failure). Interestingly, while GM positivity predicted mortality, an isolated positive *Aspergillus* PCR did not [16]. Molecular tools seem promising, especially in the improvement of our ability to diagnose IM, whose diagnosis is based on direct examination using low sensitivity special stains and culture [17]. The MODIMUCOR study evaluated a quantitative PCR (qPCR) assay, targeting *Rhizomucor*, *Lichtheimia* and *Mucor*/*Rhizopus*, performed in serum twice weekly in 232 and 13 patients, respectively, with the suspicion of invasive mold disease (IMD) (cohort 1) and proven/probable IM (cohort 2). Overall, 40 cases of proven/probable IM were diagnosed (cohort 1 *n* = 27, cohort 2 *n* = 13; overall, 67% HM and 52% grade IV neutropenia). In cohort 1, 23/27 patients with IM had a positive qPCR (serum sampling was suboptimal for four patients with negative qPCR); furthermore, an additional 18 patients had a positive qPCR (PCR-only group). Overall, the sensitivity was 85% and the specificity was about 90%, with a NPV of almost 98%. Among proven/probable IM cases, 25/40 patients had *Mucorales*-positive cultures and more fungal identifications were achieved directly from tissues by molecular tests; 36/40 had a positive serum qPCR that was 100% in accordance with the species identified from tissue or BAL. The first positive qPCR was observed at a median of 4 days before the positive mycological or histological specimen. Survival at 30 days and 6 months was significantly higher among patients whose qPCR results became negative within 7 days after amphotericin-B initiation than among patients for whom qPCR remained positive, both in the proven/probable IM and the *Mucorales* PCR-only groups [18]. Currently, there are three commercial PCR assays for *Mucorales*: MucorGenius (PathoNostics), MycoGenie *Aspergillus*-*Mucorales* (Ademtech) and Fungiplex (Bruker) [17]. The MucorGenius, which amplifies 28S ribosomal RNA of *Rhizopus*, *Mucor*, *Rhizomucor*, *Lichtheimia* and *Cunninghamella*, is the one with more evaluable data and good sensitivity, but it may be less efficient at detecting *Lichtheimia corymbifera* [19,20]. Its clinical relevance, in comparison to in-house assays and conventional mycology, was evaluated in 319 immunocompromised patients (42% HM, 11% allo-HSCT, 19% SOT) with proven/probable IM (*n* = 6), proven/probable IA (*n* = 63), *Aspergillus*–*Mucorales* co-infections (*n* = 4), possible IMD (*n* = 152) and no IMD (*n* = 94). In proven/probable IM, the in-house and MucorGenius assays showed a sensitivity of 100% and 90% and a specificity of 95.7% and 97.9%, respectively. Moreover, MucorGenius allowed the detection of *Mucorales* DNA in samples from 10 possible IMDs, all missed by culture [21]. The plasma mcfDNA-Seq also displays a high specificity, NPV and PPV for IMD, with a moderate sensitivity. It was evaluated in HSCT recipients with pneumonia (*n* = 51 proven/probable IA, *n* = 24 proven/probable non-*Aspergillus* IMD, *n* = 20 possible IMD, *n* = 19 non-IMD controls); among proven/probable IMD, it showed a sensitivity of 51%, which increased to 61% in samples obtained within <3 days of diagnosis and 79% in non-*Aspergillus* IMD, and was negative in non-IMD controls [22]. Another study on HSCT recipients with pulmonary IMDs (16 IA, 23 non-*Aspergillus* IMDs) showed that mcfDNA-Seq detected ≥1 pathogenic molds in 38%, 26%, 11% and 0% of plasma samples collected during the first, second, third, and fourth week before clinical diagnosis, respectively. Additionally, in non-*Aspergillus* IMDs, plasma mcfDNA concentrations may correlate with extra-pulmonary spread and mortality at the 6 week mark [23].

Moving to imaging techniques, a RCT highlighted [¹⁸F]FDG-PET computed tomography (CT) as a valid tool to support decision-making regarding antibiotics cessation or de-escalation in leukemic patients with persistent or recurrent NF, after induction chemotherapy or allo-HSCT, almost all of whom (90%) were receiving MAP. Patients were randomly assigned to [¹⁸F]FDG-PET-CT (*n* = 65) or conventional CT (*n* = 69); the primary endpoint was a composite of starting, stopping or changing the spectrum of antimicrobials within 96 h. Antimicrobial rationalization occurred in 82% of patients in the [¹⁸F] FDG-PET-CT group and 65% in the CT group (*p* = 0.03), mainly with the de-escalation of carbapenems and glycopeptides. There were six and four cases of proven/probable IFDs in the [¹⁸F]FDG-PET-CT and CT groups, respectively; while zero and five cases of possible IFDs occurred in the [¹⁸F]FDG-PET-CT and CT groups, respectively. Due to its metabolic component, [¹⁸F]FDG-PET-CT seemed more accurate in the identification of true infection than conventional CT, which identified many pneumonias-pneumonitis in which no causative pathogen was identified, and which might never have represented infection [24].

Finally, it is of paramount importance to correctly define b-IFDs. Thus, consensus definitions of persistent, refractory, relapsed, and b-IFDs were developed, to facilitate epidemiologic analysis and comparisons between clinical studies [25]:➢Breakthrough IFD: an IFD occurs during exposure to an antifungal, irrespective of the treatment intention (prophylactic, empiric, pre-emptive, targeted), including fungi outside the drug’s spectrum of activity. The time-point of b-IFD is the first attributable clinical sign/symptom, mycological findings or radiological feature. As a pre-existing IFD is not a b-IFD, b-IFDs can be diagnosed only if the time-point of b-IFD occurs after a minimum antifungal exposure, according to drugs’ pharmacokinetic and pharmaco-dynamic properties (time to steady state: Anidulafungin, 1 day; Caspofungin, 4–7 days; Micafungin, 4–5 days; Fluconazole, 5–10 days without a loading dose; Voriconazole, 1 day with a loading dose, 5 days without a loading dose; Posaconazole, 3–7 days; Isavuconazole, 4–7 days with a loading dose, 10–14 days without a loading dose; liposomal Amphotericin-B, 4–7 days; Olorofim, 1–2 days).➢Relapsed IFD: the IFD was caused by the same pathogen at the same site, occurring after AFT discontinuation. For probable IFD, the same clinical picture (imaging results, an increase in non-culture-based fungal biomarkers like GM in case of IA) is sufficient. A differentiation between relapse and flare during immune reconstitution is essential.➢Refractory IFD: an IFD with worsening or new attributable clinical signs, or radiological findings attributable to IFD while on treatment. Immune reconstitution can complicate assessment, as it may also lead to radiological and clinical progression that coincides with immune system recovery, so it needs to be ruled out.➢Persistent IFD: IFD unchanged from baseline, may precede treatment success. Persistent IFDs may vary by patient group (e.g., persistent disease may represent a therapeutic response in persistently neutropenic hosts, while it could represent a lack of response in patients who have become more immune-competent during the course of IFD).

## 5. Invasive Fungal Diseases in Patients Undergoing Remission Induction Chemotherapy for Acute Leukaemia

For AML patients receiving induction chemotherapy, the concomitant presence of multiple predisposing factors usually concurs to IFD development [26,27,28,29,30,31]. These risk factors can be divided into four main categories: leukemia-related (an advanced stage of the disease, unfavorable genetic pattern or a failure to obtain complete remission (CR)), host-related (performance status, comorbidities, older age or organ dysfunction), treatment-related (deep and prolonged neutropenia or mucositis) and fungal exposure-related factors (rooms without HEPA filters or previous IFDs) [1,32]. It is challenging to provide precise estimates of the incidence and etiology of IFDs, as they change over time and differ greatly among countries. This emphasizes the need for local epidemiological data and differences in at-risk patient populations, AFP practices and environmental exposure [4,26,31,33].

In 2007, a RCT with posaconazole in patients with AML undergoing induction chemotherapy resulted in 2% proven/probable IFDs, compared to 8% with fluconazole or itraconazole, translating to an OS benefit and leading to FDA approval for IFD prevention [34]. This and other subsequent studies [35,36,37,38,39] placed a large emphasis on the role of MAP in this setting and established posaconazole as the preferred agent for primary AFP undern international guidelines [7,8,9,10,40]. Yeasts, and especially *Candida* spp., have historically been the most frequent cause of IFDs [41]. However, recent epidemiological studies clearly demonstrated that IC rarely occurs in HM patients, probably due to effective AFP [42]. In contrast, molds are the leading cause of IFDs nowadays, with a major role of *Aspergillus* spp.; *Zygomycetes*, *Fusarium* and other molds are reported, but less frequently involved [26,31,33]. In addition, there has been a steady increase in IFDs caused by non-*albicans Candida* species and non-*Aspergillus* molds, which could be directly related to the extensive use of AFP with azoles [43]. These epidemiological shifts are reported in Table 1, which summarizes the recent real-life cohorts describing b-IFD’s incidence in patients undergoing induction, consolidation and/or salvage chemotherapy.

In the setting of remission induction chemotherapy, the incidence of b-IFDs in patients receiving AFP, mainly with posaconazole, was between 2.2% and 5% [44,45,46,47,48]. Interestingly, the only study reporting a higher incidence (7.9%) still employed oral suspension, leading to low posaconazole plasma concentrations (PPCs) [49]. Other mold-active agents (e.g., voriconazole, isavuconazole, echinocandins) have been used with variable results. Higher rates of b-IFDs ranging between 5.8% and 8.3% were reported in AML patients receiving AFP with isavuconazole, despite fewer drug–drug interactions (DDIs) [50,51,52]. Retrospective analyses reported an incidence for b-IFDs of around 3% in patients with HM receiving AFP with voriconazole [53,54]. Finally, AFP with echinocandin resulted in a high rate of b-IFDs in some studies [55,56], but not in others [57,58,59]. In addition, studies evaluating AFP effectiveness were historically focused on induction chemotherapy, with few ones exploring the role of AFP, especially MAP, in consolidation chemotherapy, a setting at lower risk of IFDs [26,60,61,62,63,64]. Furthermore, IFDs still constitute a major cause of expenditure in HM management [65]; therefore the prevention of IFDs represents nowadays a major priority.

Finally, the treatment landscape for AML has recently shifted to molecular targeted therapies, such as BLC2-inhibitors (venetoclax), FLT3 inhibitors (midostaurin, gilteritinib, quizartinib and sorafenib), CD33-directed antibodies (gemtuzumab ozogamicin) and IDH inhibitors (ivosidenib and enasidenib), rather than or in addition to cytotoxic chemotherapeutics or hypomethylating agents (HMAs). These targeted therapies led to an improvement in OS, with a higher achievement of CR in subgroups of patients with historically poor outcomes [66,67,68,69,70,71,72,73,74]. The incidence of IFDs with these new therapies is largely variable [55,75,76], and few data are available on the need for and efficacy of AFP. In addition, DDIs between azoles and these new molecules, mainly driven by the strong inhibition of cytochrome P450 3A4 [77], make the choice of AFP even more difficult and heterogeneous between centers. In the case of co-administration of venetoclax and triazoles, it is recommended to reduce the venetoclax dose by 75% and 50% when co-administered with posaconazole and isavuconazole, respectively; conversely, HMAs (azacitidine or decitabine) and midostaurin require no dose alterations while monitoring for eventual toxicities [77,78]. Table 1 includes studies on patients receiving both midostaurin [55] or HMAs; in the latter setting, the incidence of IFDs ranged between 3% and 12.6%, with a significant variability of AFP and a high percentage of patients receiving echinocandins or fluconazole [46,57,75,79].

## 6. Invasive Fungal Diseases in Patients Undergoing Allogeneic Hematopoietic Stem Cell Transpantation

In 2007, a RCT of allo-HSCT recipients with a-GVHD proved the superiority of posaconazole compared to fluconazole, in preventing proven/probable IA (2.3% vs. 7.0%) and b-IAs (1.0% vs. 5.9%) and reducing IFD-related mortality (1% vs. 4%) [80]. While there is international consensus on the indication for posaconazole AFP in the a-GVHD phase, in the pre-engraftment phase there is more heterogeneity on the use of AFP. In 2015, Bow and colleagues performed a mixed treatment comparison meta-analysis, to compare RCTs examining the use of oral drugs for AFP in allo-HSCT recipients. Five RCTs were included with 2147 patients, and it was found that, compared to fluconazole, itraconazole (OR 0.52 (0.35–0.76)), posaconazole (OR 0.56 (0.32–0.99)), and voriconazole (OR 0.46 (0.28–0.73)) reduced the incidence of proven/probable IFDs. Moreover, posaconazole (OR 0.31 (0.17–0.58)) and voriconazole (OR 0.33 (0.17–0.58)) reduced proven/probable IA compared to itraconazole (OR 0.68 (0.42–1.12)), while all-cause mortality was similar across all mold-active agents [81]. However, the included studies were heterogeneous in terms of study design, patient population and risk of IFDs, thus limiting the interpretations of the treatment effects, due to those biases. Winston and colleagues observed the highest IFD rate, but a greater proportion of fluconazole compared to itraconazole recipients received an unrelated donor transplant and experienced acute and chronic GVHD [82]. The percentage of patients receiving myeloablative conditioning (MAC) was different across the studies (100% [83,84], 78% [82] and 58% [85]), thus influencing the pre-engraftment myelosuppression and cytotoxic therapy-induced intestinal damage. Other significant variances among the studies were the AFP initiation (beginning of conditioning [83], the day of transplant [84,85], the day after transplant [82], the day of GVHD onset [80]) and the incidence of an a-GVHD grade ≥ two (100% [80], 64% [83], 46% [85], 41% [84] and 37% [82]). Finally, there were also significant variations in toxicity or intolerance-driven drug withdrawal rates, and the percentage of transplant from mismatched/unrelated donors was limited, not exceeding 41–48% [83,84,85]. Nowadays, the main criteria that guide clinicians to set up MAP in the pre-engraftment and early post-engraftment phases are the following: active disease at transplant, cord blood transplant, previous allo-HSCT, an a-GVHD grade ≥ 2, a mismatched related donor (MMRD), a matched or mismatched unrelated donor (MUD or MMUD), grade IV neutropenia before allo-HSCT or *Cytomegalovirus* reactivation within 100 days [12].

In the past decade, further studies on IFDs and AFP practice in allo-HSCT recipients have been published (Table 2) [2,52,86,87,88,89,90,91,92,93,94,95,96,97].

An Austrian study on recipients at low–moderate risk of IFDs (59% of whom had reduced intensity conditioning, 80% of whom had a matched donor), with only 43% receiving AFP during the pre-engraftment phase, reported that 11.6% of patients developed IFDs displaying a significantly higher mortality (48% vs. 25%) [86]. The Swiss cohort (5% of whom had had prior transplant, 23% of whom had a mismatched donor) reported a cumulative incidence of proven/probable IMD after 1 year of 7%, of which 69% were b-IMDs; overall, 29% of IMDs occurred in the pre-engraftment phase, with most patients receiving fluconazole, and the all-cause 1-year mortality rate was 48% and 31% in patients with and without IMD [87]. Moving to studies with a universal use of MAP with azoles, the cumulative incidence of IFDs did not exceed 3.2% [88,89]. Two papers described a cohort of 95 recipients receiving AFP with isavuconazole, reporting a favorable safety profile (premature discontinuation: 15% isavuconazole vs. 44% voriconazole) [88], but a trend towards a higher incidence of b-IFDs (mostly candidemia) [88,90]. Two studies evaluated a tailored AFP strategy according to IFD risk: high-risk patients received posaconazole in the study by Malagola and micafungin in the study by Busca, while fluconazole was administered to low-risk patients. The cumulative incidence of proven/probable IFDs ranged from 9.6% to 18% [91,92].

Among studies not entirely focused on allo-HSCT recipients, two analyzed the effectiveness of posaconazole (oral suspension) and isavuconazole AFP. In the first study, allo-HSCT recipients with a-GVHD represented 35% of the population: 3.2% of patients (9/270) developed IFDs (1/9 allo-HSCT recipients), a sub-optimal exposure to posaconazole was frequent and more than one third of deaths were caused by IFDs [94]. Considering the study on isavuconazole, allo-HSCT recipients were 43% of the sample; overall, 7.6% of patients (11/145) developed proven/probable b-IFDs (1/11 allo-HSCT recipients) with more than half dead at 6 weeks [52]. Another study focused on candidemia in HM patients: 133 episodes were analyzed, 42% of which were breakthrough infections while on AFP (more frequently *C. krusei* or fluconazole-resistant isolates), and the 30-day mortality was considerable (36%, 25/48 deaths were classified as *Candida*-related) [2]. A Spanish study enrolled HM patients diagnosed with b-IFDs, reporting 121 cases (42% of which were allo-HSCT recipients); among microbiologically documented b-IFDs, any fungus susceptible to the prior antifungal administered at good therapeutic levels was isolated (except some catheter-related fungemia). Again, the mortality after 100 days was high (47%, with 61% of those deaths caused by b-IFD), especially in mucormycosis and other rare yeasts. Interestingly, the molecular testing of biopsy samples identified a high number of non-*Aspergillus* molds [95]. Lastly, two studies analyzed b-IFDs in immunocompromised patients receiving MAP with azoles, and the incidence of proven/probable IFDs was 4.9% [96] and 6.3% [97].

Another important issue is the change in transplant platforms, including the advent of post-transplant cyclophosphamide (PT-Cy) to allow transplantation from alternative donors, especially haploidentical donors, while modulating the consistent risk of GHVD. The rate and characteristics of infectious complications with PT-Cy have been reported in different studies, but few were dedicated specifically to IFDs [98]. A retrospective study (2011–2021) analyzed 205 haploidentical HSCT recipients receiving PT-Cy, mainly after reduced intensity conditioning (75%) and without the routine administration of AFP. Overall, 29 patients (14%) developed proven/probable IFDs (14/29 within 30 days post-HSCT (13/14 invasive yeast infections), 15/29 after 30 days post-HSCT (7/15 IMDs, of which 6/7 were IA, 6/15 were invasive yeast infections, 1 was pneumocystosis and 1 was coccidioidomycosis)). The 30-day cumulative incidence of proven/probable invasive yeast infection was 6.3%, with significantly increased non-relapse mortality occurring in these patients (54% vs. 11%) [99]. Another retrospective study (2013–2015) investigated 381 haploidentical HSCT recipients receiving PT-CY (38% of whom had had a prior transplant): IFDs were reported in 78 patients (mostly caused by *Aspergillus* (43%) and *Candida* (33%)), the rate of IFDs was 6% and IFDs occurred at a median of 20 days post-HSCT. Notably, MAP with posaconazole was provided only during GVHD, which might have contributed to cases of early IA; finally, among 37 patients who died of infection-related mortality, only 4/37 deaths were caused by IFDs [100]. In a Spanish multicenter study (2013–2018) on 236 PT-CY haploidentical HSCT recipients, the incidence of IFDs within 30 days post-HSCT was 4%, reaching 11% within 3 years; the most frequent IFD was proven/probable IA (*n* = 16), followed by candidiasis (*n* = 16) and pneumocystosis (*n* = 2). In this cohort, AFP regimens differed between centers, with some using fluconazole and others MAP [101]. Finally, the CAESAR study, which developed a clinical risk score for predicting IFDs within 6 months from allo-HSCT, reported haploidentical HSCT as a risk factor [102]. In conclusion, haploidentical HSCT with PT-Cy seems to be associated with a high risk of IFDs, particularly early IFDs, and dedicated prophylactic strategies might be warranted.

## 7. Single-Center Experience with Posaconazole Prophylaxis in Patients with Acute Myeloid Leukaemia

We retrospectively reviewed the medical records of 662 adults diagnosed with AML/MDS at San Raffaele Hospital between 2010 and June 2023, undergoing induction, consolidation and/or salvage chemotherapy. We excluded patients who did not receive intention-to-cure treatment, acute promyelocytic leukemia, subjects receiving secondary AFP for pre-existing IFD and those who received AFP other than posaconazole. Thus, we analyzed the data of 184 patients (totaling 339 chemotherapy treatments).

### 7.1. Methods, Objectives and Statistical Analysis

Patients gave written informed consent for chemotherapy and the use of their medical records for research and were treated according to the institutional standard of care. Baseline patients’ characteristics, hematological disease features, details about chemotherapy cycles and hematological response assessments were analyzed. Data were anonymized and managed according to the Good Clinical Practice guidelines published by the World Medical Association’s Declaration of Helsinki. The analyses were performed considering the hematological treatments’ episodes: remission induction chemotherapy, consolidation chemotherapy and salvage chemotherapy. Hematological treatment platforms were uniform throughout the study period; conversely, anti-leukemia targeted therapies were progressively introduced over time.

The primary objective was to estimate the incidence rates of proven/probable b-IFDs; the secondary objectives included the description of characteristics and outcomes of patients who experienced proven/probable b-IFDs.

Baseline (day 0) was defined as the date of chemotherapy cycle initiation; the follow-up of each hematological treatment episode was censored at patient’s death, subsequent eventual chemotherapy cycle or HSCT, or on the follow-up visit on day 180. Neutropenia was defined as ANC < 500 cells/mm^3^ [11,40]. The hematological response was assessed for induction chemotherapy cycles at day 14 to evaluate the eventual indication to early re-induction; otherwise, it was assessed at day 30. High-risk neutropenia before chemotherapy was defined as ANC < 500 cells/mm^3^ at baseline, for at least 7 days. Engraftment was defined as the first of three consecutive days with ANC > 500 cells/mm^3^ [11,40]. Proven/probable b-IFDs were defined according to EORTC/MSG definitions, and b-IFD diagnosis was dated at the time of symptoms’ onset, radiological signs or mycological evidence of IFDs; whichever occurred first [25]. Posaconazole was started at baseline. PPCs were assessed weekly and they were considered inadequate if <0.7 mg/L, adequate for prophylaxis if they measured 0.7–1 mg/L and adequate for both prophylaxis and treatment if >1 mg/L [7,8]. The clinical outcome of patients with b-IFD was assessed at 6 and 12 weeks after diagnosis.

Incidence rates (IRs) of proven/probable b-IFDs in each chemotherapy category were calculated with univariable Poisson regressions; rates per 100 person years follow-up (PYFU) were reported; 95% confidence intervals (95% CI) were estimated with a normal approximation to binomial. Two-sided *p*-values < 0.05 were considered significant. Incidence rate ratios (IRRs) were calculated to compare IRs. Statistical analyses were performed using R (R Foundation for Statistical Computing, Vienna, Austria, version 4.2.3).

### 7.2. Results

Overall, 339 chemotherapy treatments were analyzed from 184 patients (a patient may have received more than one chemotherapy treatment): 150 patients received 153 remission induction chemotherapy treatments, 90 patients underwent 126 consolidation chemotherapy treatments and 54 patients received 60 salvage chemotherapies. The total number of treatment cycles analyzed was 1052 (1 to 12 maximum cycles per patient). The characteristics of these chemotherapy treatments are reported in Table 3.

Overall, the median age at chemotherapy initiation was 61.5 years (IQR 44.6–69.5). Among induction chemotherapy treatments, 83% (127/153) were chemotherapy-based and 17% (26/153) were HMAs; conversely, among consolidation chemotherapy treatments, 87 high-dose cytarabine and 27 intermediate-dose cytarabine treatments were administered to 62 and 22 patients, respectively, while 9.5% (12/126) of treatments were based on HMAs. Salvage chemotherapy treatments included chemotherapy-based regimens in 72% of cases (43/60), HMAs in 15% (9/60) and chemotherapy-free regimens based of target drugs in 13% (8/60). Finally, 8.5% (29/339) of treatments included new target drugs, distributed as illustrated in Table 3. Overall, high-risk neutropenia was present in 25.7% of cases (86/339) before treatment initiation, especially in induction and salvage chemotherapy groups.

Six cases of probable/proven b-IFDs were recorded in six patients. Overall, the IR of proven/probable b-IFDs was 1.7% and, when categorized according to the type of chemotherapy treatments, it was 1.3% for induction, 0.8% for consolidation and 5% for salvage chemotherapy, respectively (Table 3). The IRs of b-IFDs per 100 PYFU was 5.04 (0.47, 14.45) in induction chemotherapy (*n* = 2), 3.25 (0.0013, 12.76) in consolidation (*n* = 1) and 18.38 (3.46, 45.06) in salvage chemotherapy (*n* = 3). There were no significant differences in b-IFDs IRs between induction and consolidation chemotherapy (IRR 1.55 (0.14–17.08), *p* = 0.72), consolidation and salvage chemotherapy (IRR 0.18 (0.02–1.70), *p* = 0.09) and induction and salvage chemotherapy (IRR 0.27 (0.04, 1.64), *p* = 0.13). All b-IFDs were probable pulmonary IA (mycological criteria: 6/6 GM positivity (no positive cultures)) and are detailed in Table 4.

Three probable IA occurred in patients with adequate PPCs at the time of diagnosis, while two patients were under-exposed to posaconazole; most patients (5/6) received voriconazole as AFT. At 12 weeks post-b-IFD diagnosis, treatment success was achieved in all cases. Moreover, no deaths were attributable to b-IFDs and all patients regularly proceeded in their course of leukemia treatments. The neutrophils engraftment was achieved in 86.3% of treatments, although with different percentages among groups (87.3% in induction, 94.4% in consolidation and 66.7% in salvage chemotherapy; *p* < 0.001). Similarly, a complete response to treatment occurred in 78.7% of cases (248/339), with significantly different proportions between treatments groups (78.7% in induction, 89.1% in consolidation and 56.4% in salvage chemotherapy; *p* < 0.001). Overall, 19% of patients (35/184) died during follow-up, more frequently in the salvage chemotherapy group (29.6%) compared to the induction (7.3%) and consolidation chemotherapy (8.9%) groups. The most common cause of death in the induction and salvage chemotherapy groups was relapsed refractory disease, while in the consolidation chemotherapy group half of the deaths were due to bacterial infections.

Overall, PPCs were available for 95.1% of patients (175/184) and 95.3% of chemotherapy treatments (323/339), with a total of 936 measurements (0 to 19 maximum measurements per patient). Table 3 reports details about PPC measurements and levels, according to different chemotherapy treatment groups; interestingly, at least two consecutive inadequate PPCs were recorded in only 6.8% of chemotherapy treatments (22/323).

In our cohort, the proportion of patients who developed b-IFDs during induction chemotherapy was 1.3% (2/150), one of the lowest among similar studies, even considering those that tested isavuconazole [50,51,52], thus, supporting the efficacy of posaconazole AFP in this context. Moving to consolidation chemotherapy, the proportion of our patients experiencing b-IFDs was 1.1% (1/90), lower than in studies conducted before extensive AFP use [26,60,61,62,63] and lower than in the recent study by Del Principe [64], where AFP was not provided for 44% of patients and the b-IFD incidence among those receiving MAP was 1.5%. Notably, two out of six probable IA in our cohort occurred in patients treated with VEN–HMAs, with 10.5% of patients receiving VEN–HMAs and experiencing b-IFD (2/19), which is similar to the proportion reported in one study [57] but higher than in other cohorts [46,75,76,79]. In this setting, some studies [46,57] suggested AFP during initial cycles of induction and in relapsed/refractory AML, while other studies [75,76,79] stated there was no need for AFP. The addition of VEN to HMAs improves OS and rates of CR, compared to HMAs alone, but is associated with prolonged neutropenia [66]. The role of AFP in patients receiving VEN–HMAs deserves careful consideration, at least until further studies are available. Analyzing the outcomes of our patients with b-IFDs, at week 12 all subjects achieved treatment success and continued their hematological treatments. These findings are appreciable, thus suggesting that a broad-spectrum AFP may reduce the burden of IFDs, resulting in less severe forms. Previous studies reported a similar positive effect of AFP on the effectiveness of AFT [64,103]. Even if our first-line AFT did not comply with most guidelines, by favoring an antifungal of a different class [104,105,106,107,108], the treatment success was favorable, underlying how the optimal management of b-IFDs is largely unknown and the clinical studies investigating different treatment modalities are scarce [106,109,110,111]. To the best of our knowledge, our study is the one with the largest PPC measurements. Overall, in 20.1% of chemotherapy treatments, the PPCs totaled <0.7 mg/L at one time-point (only 6.5% at two consecutive time-points), and this percentage is lower than that reported by several studies [44,49,112], possibly due to a more extensive use of tablet formulations, but still indicates considerable interpatient variability. Although there are concerns about the suboptimal efficacy of azoles in the context of neutropenia [113], the importance of achieving PPCs > 0.7 mg/L for prophylaxis and >1 mg/L for treatment [7,8,114], and the absence of a definite correlation between PPCs and toxicity, are well known [114,115]. A recent meta-analysis indicated a C_min_ of 0.5 mg/L as the cut-off that separates successful from failed prophylaxis [116]. These data confirm the importance of performing TDM to optimize MAP and treatment.

Our retrospective study has several limitations. Firstly, the low number of events did not allow us to investigate predictive factors for b-IFDs, or to draw conclusions regarding b-IFD management and clinical outcomes, especially IFD-attributable mortality, which was remarkably low in our cohort. Since it was not designed as a case–control study, we did not compare different AFP strategies. We also did not collect medical conditions potentially affecting posaconazole pharmacokinetics. Our cohort included a low percentage of patients treated with HMAs (8.3%), VEN–HMAs (5.6%) and target therapies (8.5%), so it is unfeasible to draw meaningful conclusions about the incidence of b-IFDs in these novel settings of AML treatment, considering that larger studies have been published (Table 1).

Finally, an effective containment of IFD occurrence in patients undergoing induction chemotherapy, as emerged in our cohort, is fundamental, because several patients will be candidates for allo-HSCT, and a history of previous IFDs is a risk factor for relapsing IFDs after transplantation.

## 8. Final Remarks and Future Perspectives on Novel Antifungal Drugs

IFDs represent one of the causes of persistent NF, alongside multi-drug resistant bacterial infections, viral infections and other non-infectious reasons. At least 15% of patients with high-risk neutropenia develop recurrent NF, mainly caused by microbiologically documented infections (31%) and clinically documented infections (30%), with IFDs representing almost half of microbiologically documented infections; notably, the risk of death is higher among patients with recurrent NF (5.4% vs. 1.4%) [117].

### 8.1. Practice Points

Early b-IFDs on MAP with azoles occur in AML patients during induction chemotherapy due to a high fungal inoculum exposure, previous colonization or sub-optimal pharmacokinetics of triazoles; late b-IFDs, caused by *Aspergillus* and non-*Aspergillus* IMDs, are typical of relapsed refractory AML or prolonged corticosteroid treatment for GVHD. In the absence of RCTs to guide therapeutic decisions in the different scenarios of b-IMDs, a rational AFT is liposomal amphotericin-B eventually associated with isavuconazole or posaconazole, while trying to obtain a microbiological diagnosis [107];The incidence of IC in HM patients has decreased with AFP, leading to an increased proportion of non-*albicans* Candida with decreased azole susceptibility (*C. glabrata*, *C. krusei*); among patients receiving AFP with azoles, the prevalence of candidemia and hepato-splenic candidiasis is estimated around 20% and 3%, respectively, and the emergence of echinocandin resistance (*C. glabrata*) and novel pathogenic species (*C. auris*) is an issue, although the prevalence is low [2,118];The epidemiology of b-IMDs in patients receiving AFP with azoles shows a shift towards non-*fumigatus Aspergillus* (i.e., *A. ustus* complex, usually associated with extra-pulmonary infection and high mortality >50%) [119], non-*Aspergillus* molds (*Fusarium*, *Mucorales*, *Scedosporium apiospermum*) and rare resistant molds [120];It is of paramount importance to pursue a mycological diagnosis of b-IFDs in high-risk NF patients; *Aspergillus* real-time PCR assays on blood and BAL have been included in EORTC/MSG 2020 criteria [15] and novel diagnostic tools are promising (i.e., Septifast [121], plasma mcfDNA sequencing [22,23] and real-time PCR-based methods for *Mucorales* [18,19,20] or IA sustained by mixtures of triazole-resistant and susceptible strains of *A. fumigatus* [16,122]);Exposure to mold-active agents could select already resistant *Aspergillus* strains (i.e., cryptic species) and non-*Aspergillus* molds, rather than promote the induction of resistance “in vivo”; not only the molecular identification of non-*Aspergillus* molds, but also that of cryptic *Aspergillus* species, whose frequency stands at 11–15%, could be clinically relevant, since antifungal drug resistance is common (40%) [123,124];Antifungal susceptibility testing is recommended for *Candida* spp. (azoles, echinocandins) and *A. fumigatus* (azoles) because the mechanisms of resistance are well-described (acquired mutations: for azoles/*Candida* mutations in the ERG11 target gene, the overexpression of target genes and/or drug transporters; for echinocandins/*Candida* FKS hot-spot mutations; for azoles/*A. fumigatus,* cyp51A gene mutation) and the performance of clinical break-points is acceptable for their identification with a high probability of treatment failure (clinical break-points to predict clinical success (CLSI for *Candida*, EUCAST for *Aspergillus*): echinocandins/*C. glabrata* MIC ≤0.25–0.5 for caspofungin, ≤0.12 for anidulafungin; azoles/*A. fumigatus* MIC ≤2 mg/L for voriconazole, ≤0.25 mg/L for posaconazole, ≤2 mg/L for isavuconazole); conversely, for non-*Aspergillus* molds, antifungal susceptibility testing should be interpreted with caution, because MIC are unreliable predictors of outcome [125];Guidance on imaging for pulmonary IA and IM exists [15,126]; high resolution lung CT is fundamental in the diagnostic work-up of IFDs; although not pathognomonic, in patients with HM and neutropenia, the halo sign is present at the symptoms’ onset in >70% of IA, but its prevalence decreases rapidly over time (<40% after 7 days, <20% after 14 days). The air-crescent sign is not specific and occurs in later stages of IFD; the reversed halo sign and hypodense sign are typical of pulmonary IM but occur less frequently.

### 8.2. Future Considerations

The optimal management of AFP, including the necessity, type and dosage of antifungals, in the novel treatment landscape of AML based on molecular targeted therapies;The development and clinical application of optimal risk stratification strategies, to prevent IFDs in allo-HSCT recipients, through an individualized choice of AFP (MAP vs. non MAP) according to patients’ features (their hematopoietic comorbidity index, disease status at HSCT, whether they have undergone previous allo-HSCT, whether they have had previous IFDs), the type of transplant (conditioning, donor (haploidentical, cord blood), GVHD prophylaxis (PT-Cy)), and an individualized AFP length (engraftment, poor graft function or a-GVHD). Early diagnosis strategies based on local prevalence, risk factors and phase after HSCT, are warranted;Prospective studies should include careful diagnosis of proven/probable IFDs and b-IFDs and correct detailing of the outcomes of IFDs, to discern the exact burden of IFDs, b-IFDs and the attributable mortality from the role of other factors (i.e., non-fungal infections, neutrophils recovery, underlying disease response, GVHD and persistent immune-suppression).

Finally, because of some limitations of current antifungals (e.g., route of administration, toxicity or DDIs) and the emergence of resistant fungi, there is still an urgent need to expand our armamentarium and several compounds are reaching the stage of phase II or III clinical assessment. These include new drugs within the existing antifungal classes or displaying a similar mechanism of activity with improved pharmacologic properties (rezafungin and ibrexafungerp) or first-in-class drugs (olorofim and fosmanogepix). Rezafungin, a long acting intravenous echinocandin with a weekly schedule of administration, has recently been FDA-approved for the treatment of IC, after the results of the STRIVE and ReSTORE trials (with the comparator caspofungin) [127,128]. Its spectrum includes *Candida* spp. (including azole-resistant strains), *Aspergillus* spp. (including azole-resistant isolates) and *P. jirovecii*, but it does not cover non-*Aspergillus* molds. Pooling the data from the two trials (139 patients for rezafungin, 155 patients for caspofungin), the 30-day all-cause mortality rates were comparable between groups (19% rezafungin, 19% caspofungin), and mycological eradication occurred by day 5 in 73% and 65% of patients treated with rezafungin and caspofungin, respectively [129]. Rezafungin is currently also being studied for prophylaxis; the phase III ReSPECT trial is ongoing to compare rezafungin vs. fluconazole as AFP in allo-HSCT recipients (drug exposure: from transplant until week 14), while considering some conditions at high risk of IFDs opt be exclusion criteria (active disease at transplant, cord blood transplant or previous allo-HSCT). Olorofim is the first compound from the novel orotomide class, which selectively inhibits fungal dihydroorotate dehydrogenase, a key enzyme in pyrimidine biosynthesis that results in the loss of essential substrates for cell wall integrity and DNA replication. It can be administered both intravenously and orally, the therapeutic concentrations are achieved in all targeted organs and tissues (including lung, brain, liver and kidney) and it is metabolized by several cytochrome P450 isoenzymes, but DDIs may be less clinically relevant than those observed with mold-active triazoles. Its spectrum includes wild-type and azole-resistant *Aspergillus* spp. and some uncommon fungi (*Scedosporium* spp., *L. prolificans*, *Fusarium* spp.), while it is inactive against *Candida* and *Mucorales* [130]. Both the phase II open label single-arm trial (NCT03583164) evaluating olorofim for IMDs with limited therapeutic options, and the phase III OASIS trial (NCT05101187) comparing olorofim vs. liposomal amphotericin B for proven/probable IA refractory to azoles, or breakthrough on MAP with azoles, or in the case of DDIs that make azoles unsuitable, are recruiting. Fosmanogepix is a first-in-class antifungal agent inhibiting Gwt1, an enzyme in the glycosylphosphatidylinositol biosynthesis pathway. It can be administered both intravenously and by the oral route, and it distributes in many difficult-to-treat body sites (including the brain, eyes and abdominal abscesses). It is highly active against *Candida* spp. (including *C. auris*, but not *C. krusei* and *C. kefyr*) and the majority of echinocandin and azole-resistant *Candida*. Regarding molds, fosmanogepix displays potent activity against *Aspergillus*, including azole-resistant *A. fumigatus* and cryptic species, and it is also active against most *Fusarium*, in particular *F. solani*, and most *Scedosporium* and *L. prolificans* isolates. In contrast, in vitro antifungal activity against most *Mucorales* is limited [130].

In conclusion, these novel drugs are promising for the treatment of emerging resistant *Candida* and *Aspergillus* spp., as well as some notoriously refractory IMDs (disseminated fusariosis, scedosporiosis/lomentosporiosis); however, they do not seem to be suitable for the treatment of IM, which remains a major issue.

**Table 1 microorganisms-12-00117-t001:** Studies reporting the incidence of invasive fungal diseases in patients with AML treated with PSZ-AFP undergoing induction, consolidation or salvage chemotherapy *.

Reference *Country*	Study Design *Study Period* Study Population *Chemotherapy*	*Antifungal Prophylaxis* *Incidence of Invasive Fungal Diseases* Therapeutic Drug Monitoring *Type of Breakthrough Invasive Fungal Diseases*	Outcome (OS, AMR) *Risk Factors for IFD* Other Analysis/Considerations
Gangneux JP [44] *France*	Multicenter, prospective; *2016–2018;* 404 adults with AML; *RIC 81%, consolidation 11%, salvage 8%.*	*92% PSZ (91% tablets), 4% echinocandins, 3% FCZ, 1% L-AMB;* **2.2%** probable/proven b-IFDs; PPCs available for 139 patients, mean PPCs = 1.2 ± 0.9 mg/L; *Five probable IPA, two probable PCP, one proven mucormycosis, one fungemia (Saccharomyces spp.).*	Follow-up at 15 days after the end of AFP: IFD-attributable mortality was 0.7% (3/404); *Not reported;* 31.2% (126/404) patients switched from AFP to AFT (66.7% empirical, 19% pre-emptive, 14.3% targeted). Factors more frequently associated with the switch were: unfavorable cytogenetics/molecular biology prognosis, chemotherapy naïve patients, not receiving AFP loading dose, increasing duration between the start of chemotherapy or AFP initiation.
Cattaneo C [55] *Italy*	Multicenter, retrospective/prospective; *2019–2021;* 114 adults with FLT3-mutated AML; *RIC with midostaurin 100% (n* = *83 achieved CR, n* = *31 did not achieved CR);* *79/83 patients received 160 consolidation courses and 12/31 patients received re-induction.*	*Induction: 93% AFP (48% PSZ, 16% echinocandin, 21% sequential PSZ–echinocandin (PSZ from day 0 to day 7, echinocandin from midostaurin initiation));* *Re-induction: 100% AFP (25% PSZ, 50% echinocandin, 7% sequential PSZ–echinocandin, ISA 2%);* *Consolidation: 46% AFP (13% PSZ, 17% echinocandin, 4% sequential PSZ–echinocandin, ISA 4%);* **10.5%** (12/114 patients) probable/proven IFD during induction, **8.3%** (1/12 patients) probable/proven IFD during re-induction, **2.5%** (4/160 chemotherapy courses) probable/proven IFD during consolidation; Not reported; *Induction: four candidemia, seven probable IPA (one associated with sinusitis), one geotrichosis (S. capitata);* *Re-induction: one probable IPA;* *Consolidation: three probable IPA (one associated with sinusitis), one candidemia.*	8.3% (1/12) IFD-attributable mortality during induction. AML response to treatment and duration of neutropenia were the predictive factors for survival; *Age;* PSZ–AFP did not lead to a higher rate of midostaurin discontinuation; No significant differences in IFD’s incidence between the different AFP strategies during induction, although IFD were more frequent with echinocandin–AFP; OS was not influenced by IFD occurrence (22 months in patient with IFD, 16 months in patients without IFD (*p* = 0.221)); but, among patients who did not achieved AML response (CR), IFD occurrence negatively affected survival.
Chen TC [45] *China*	Monocenter, retrospective; *2005–2019;* 208 adults with newly diagnosed AML; *RIC 100%.*	*28% PSZ, 72% no AFP;* **5%** (3/58) probable/proven IFDs in PSZ group, **11%** (16/150) in no AFP group (*p* = 0.22); Not available and not performed routinely; *Fifteen probable IPA, four proven IPA.*	0.8% IFD-attributable mortality during RIC (only one death related to IPA); Median OS of 689 days in PSZ group vs. 514 days in no AFP group, causes of death were superimposable, with leukaemia as the leading cause; *First induction chemotherapy failure (HR 4.48);* PSZ discontinuation rate of 34% (half of cases due to liver toxicity); No significant difference in proven/probable IPA incidence or OS between groups (PSZ vs. no AFP).
Rausch CR [46] *USA*	Monocenter, retrospective *2016–2019;* 277 adults with newly diagnosed AML; RIC 62%, VEN-containing regimen 38%.	*100% PAP (51% PSZ, 30% VCZ, 19% ISA);* **3.5%** (6/172) probable/proven IFD in patients undergoing RIC, **4.8%** (5/105) in those receiving a VEN-containing regimen; The b-IFD incidence was **2.9%** for PCZ, **4.8%** for VCZ, and **5.7%** for ISA (*p* = 0.55); Not available; *Three probable IPA, one disseminated IA, one fungemia (S. cerevisiae), one proven pulmonary cryptococcosis, one disseminated fusariosis, four candidemia.*	Mortality of 27% (3/11) 42 days after the onset of b-IFD (two deaths in PSZ group, one death in VRC group); *Not reported;* Patients with b-IFDs had a lower rate of CR (18% vs. 66%) and ANC recovery > 1000 (64% vs. 91%); There were no significant differences in IFD’s incidence or mortality between the different AFP strategies.
Aldoss I [57] *USA*	Monocenter, retrospective; *2016–2018;* 119 AML patients (55 newly diagnosed, 64 *r*/*r*); *VEN-HMAs (87% DEC, 13% AZA).*	*79% AFP (21% PSZ, 38% micafungin, 13% ISA, 4% VCZ), 21% no AFP;* 12.6% probable/proven b-IFDs, the median time from initiating VEN–HMAs to the onset of IFDs was 107 days in responders and 66 days in non-responders to HMAs; Not available; *Seven proven IFDs: two Scedosporium spp. (one fungemia, one skin-deep tissue), three mucormycosis (one disseminated, two sinus-orbit), one IPA, one disseminated Aspergillus-Mucor coinfection;* *Eight probable IFDs: five IPA, one unspecified mold lung infection, one lung mucormycosis, one Penicillium spp. pneumonia.*	The median OS superimposable between patients who did and did not develop IFDs (225 days vs. 325 days); *Failure to obtain CR/CRi (IFD incidence: 22% vs. 6%) and VEN–HMA treatment for r/r AML, compared to newly diagnosed (IFD incidence: 19% vs. 5%);* No significant differences in IFD’s incidence or mortality between the different AFP strategies; AZA associated with a higher risk of IFD (low sample size).
Chen EC [75] *USA*	Monocenter, retrospective; *2016–2021;* 131 newly diagnosed AML patients; Frontline VEN–HMAs 100%.	*17% AFP (9% PSZ, 9% VCZ, 14% ISA, 68% FCZ), 83% no AFP;* 3% (4/131) probable/proven IFDs (0/4 b-IFDs, 4/4 no AFP); Not available for PSZ; *Three proven IFDs: two candidemia (one C. krusei, one C. parapsilosis), one Fusarium skin-soft tissue infection;* *One probable IPA.*	Median OS was superimposable in patients who did and did not receive AFP (8.1 months vs. 12.5 months); *Poor fitness (HR 4.21) and TP53 mutation (HR 3.06);* No significant differences in IFD’s incidence, OS, proportion of patients proceeding to HCT or experiencing relapse according to AFP use. Concurrent VEN dose reduction did not compromise AML outcomes.
Yang E [47] *Republic of Korea*	Monocenter, retrospective; *2016–2019;* 247 AML patients; RIC 100%.	*66% (162/247) PSZ–AFP, 34% (85/247) no AFP;* **4.9%** (12/247) proven/probable IFDs, 2.5% (4/162) proven/probable b-IFDs vs. 9.4% proven/probable IFDs (8/85) (*p* = 0.03); Not reported; *Ten mold infections (eight IA (2/8 in PSZ–AFP), two mucormycosis (1/2 in PSZ–AFP)), two invasive candidiasis (1/2 in PSZ–AFP).*	Only 1/12 of patients with proven/probable IFDs died; *Not reported;* Receiving AFP was associated with less need for AFT.
Del Principe MI [64] *Italy*	Multicenter, retrospective; *2011–2015;* 2588 AML patients in CR after RIC; Consolidation CT 100%.	*56% AFP (43% PSZ, 54% ITZ, 1% VCZ, 2% L-AMB), 44% no AFP;* **2.2%** (56/2588) probable/proven IA (1.7% probable (43/2588), 0.5% proven (13/2588)), 22/56 cases were b-IFD; Not available; *Overall, 89% (50/56) IPA, 14% (8/56) disseminated IA, 7% (4/56) had previous IFDs during RIC.*	In patients with proven/probable IA, AMR within 120 days of IFD onset was 9%; *None;* AFP decreased IFD’s incidence (1.5% and 2.9% proven/probable IA in patients who received AFP vs. those who did not, respectively (*p* < 0.01)) and promoted a response to first-line AFT; Age > 60 years (OR 12.4) and high-dose cytarabine (10.6) were predictive of AMR. AFP during RIC does not have a protective effect on IA’s incidence during consolidation CT.
Del Principe MI [103] *Italy*	Multicenter, retrospective; *2011–2015;* 1450 *r*/*r* AML patients; Salvage CT 100%.	*Not reported overall, available for 73/74 patients with proven/probable IA (80% AFP (43% PSZ, 24% FCZ, ITZ 17%, L-AMB 12%), 20% no AFP);* 5.1% probable/proven IA (0.7% (10/1450) proven (10/10 b-IFDs), 4.4% (64/1450) probable (48/64 b-IFDs)), 21% (14/74) of patients with proven/probable IA had a previous fungal infection; Not available; *Proven IA (ten): four A. fumigates, three A. niger, two A. terreus, one A. oryzae (3/10 b-IFDs during PSZ-AFP);* *Overall, serum and BAL GM positive in 54/74 (73%) and 17/74 (23%), respectively.*	In patients with proven/probable IA, the overall mortality was 48%, and AMR within 120 days of IFD onset was 27%; *Not reported;* AFP administration promoted a better response to AFT. A mucositis grade > two and AFP administration were predictive of the overall response rate (both complete and partial responses to first-line AFT).
Hsu A [48] *USA*	Monocenter, retrospective; *2007–2019;* 108 AML patients; RIC 100%.	*52% AFP (84% PSZ, 12% FCZ, 4% VCZ), 48% no AFP;* **2.8%** (3/108) proven/probable IFDs, AFP use was associated with less proven/probable IFDs (0% b-IFDs vs. 6%); Not available; *Three proven invasive candidiasis.*	AFP was associated lower all-cause in-hospital mortality (7% vs. 21%, *p* < 0.05); *Not available;* AFP was associated with less need for AFT.
Michallet [49] *France*	Multicenter, prospective; *2009–2013;* 677 patients; RIC 100%.	*30% no AFP (group-1), 36% PSZ–AFP (100% suspension; group-2), 21% PSZ–AFP plus another antifungal (caspofungin, L-AMB, FCZ; group-3), 13% other AFP (caspofungin, L-AMB, FCZ; group-4);* **9%** (61/677) proven/probable IFDs; according to AFP, the cumulative incidence of probable/proven IFDs was **13.8%** (group-1), **7.9%** (group-2), **5.6%** (group-3), and **6.6%** (group-4) at day 60; PPCs available for 92/383 patients receiving PSZ: only 21.7% PPCs > 0.7 mg/L; *Proven/probable IFDs: 38 IA, 17 invasive candidiasis, 6 invasive mucormycosis.*	After a median follow-up of 27.5 months, the mortality rate was 38.3%, with 5.4% IFD-attributable mortality; *Not reported;* AFP decreased the incidence of IFDs significantly and delayed IFD’s onset (26 days after RIC in groups 2–3, 16 days in group 1 and 20 days in group 4); There were no significant differences in the incidence of IFDs between different AFP strategies; In patients without IFDs, the rate of AML complete remission was higher than among those with IFDs (88% vs. 80%); Unfavorable cytogenetics (HR 3.3) and IFD occurrence (HR 5.6) were predictive of mortality by day 100. No b-IFD when PPCs > 0.7 mg/L.
On S [79] *USA*	Monocenter, retrospective; 2016–2020; 235 AML patients; VEN-HMAs (55% in newly diagnosed AML, 45% in *r*/*r* AML).	*67% AFP (43% PSZ, 28% VCZ, 18% ISA, 4% FCZ, 7% echinocandins), 33% no AFP;* **5.1%** (12/235) probable/proven IFDs, all 12 patients were neutropenic at IFD onset (median duration of 46 days (IQR: 2–178)), 6/12 were b-IFDs (five MAP, one FCZ); Not available; *Proven IFDs: three invasive candidiasis, one sinus and three pulmonary mold infections (two Aspergillus, two Mucor);* *Probable IFDs: one Candida esophagitis, four pulmonary mold infections (two Aspergillus).*	Not available; *Not available;* Incidence rates of probable/proven IFDs did not differ significantly between patients with newly diagnosed AML and those with *r*/*r* AML, nor between AFP and no-AFP groups.

Abbreviations: AFP, antifungal prophylaxis; AFT, antifungal therapy; AML, acute myeloid leukemia; AM, attributable mortality; AMR, attributable mortality rate; AZA, azacitidine; CT, chemotherapy; DEC, decitabine; FCZ, fluconazole; HR, hazard ratio; IFD, invasive fungal disease; IA, invasive aspergillosis; IPA, invasive pulmonary aspergillosis; ISA, isavuconazole; ITZ, itraconazole; HCT, hematopoietic stem cell transplant; HMA, hypomethylating agent; L-AMB, liposomal amphotericin B; MAP, mold-active prophylaxis; OS, overall survival; PAP, primary antifungal prophylaxis; PCP, pneumocystosis; PPCs, plasmatic posaconazole concentrations; PSZ, posaconazole; RIC, remission induction chemotherapy; VCZ, voriconazole; VEN, venetoclax. * Excluding studies published before 2018 and with a sample size below 100 patients.

**Table 2 microorganisms-12-00117-t002:** Studies reporting the incidence of invasive fungal diseases and/or the antifungal prophylaxis practice in allogeneic hematopoietic stem cell transplant recipients (eight studies entirely on transplant recipients, six studies on immune-compromised hosts including allogeneic hematopoietic stem cell transplant recipients).

Reference *Country*	Study Design *Study Period* Study Population	Antifungal Prophylaxis *Incidence of Invasive Fungal Diseases* Therapeutic Drug Monitoring *Type of Breakthrough Invasive Fungal Diseases*	Clinical Outcome *Risk Factors for Invasive Fungal Diseases* Other Analysis/Considerations
Harrison N [86] *Austria*	Monocenter, retrospective; *2009–2013;* 242 allo-HSCT recipients; Conditioning: MAC 41%, RIC 59%; Donor: 80% matched donor, 20% mismatched donor.	43% of patients (105/242) received AFP during aplasia (PSZ *n* = 52, VCZ *n* = 28, FCZ *n* = 17); ***11.6%** of patients (28/242) experienced a proven/probable IFD, 25/28 during the first year after HSCT;* *Among cases of invasive candidiasis or IA (25/28), 56% (14/25) were without AFP, 16% (4/25) b-IA on FCZ, 28% (7/25) b-IFD on MAP;* Not available; *Proven IFDs: five IC (one C. albicans, two C. glabrata, one C. krusei, one Candida spp.), three IA (one A. fumigatus, two Aspergillus spp.);* *Probable IFDs: two IC (two C. albicans), three P. jirovecii, three IA (one A. fumigatus, one A. niger, one Aspergillus spp.), twelve IA with positive galactomannan antigen.*	IFD was associated with a higher mortality rate compared to patients without IFD (48% vs. 25%, *p* = 0.02), and the higher mortality rate was even more pronounced for IA (62%, *p* = 0.003); Patients with IFD were admitted to the ICU more often (18/28, 64%) than patients without IFD (26/214, 12%); *Intensified GVHD therapy (≥1 mg/kg corticosteroids and basiliximab or etanercept) and TAM were associated with an increased risk for IFD and IA, while AFP was associated with a decreased risk;* The median time from HSCT to IFD diagnosis was 8 days for IC, 36 days for IA and 319 days for pneumocystis pneumonia; 5.3% (6/113) of patients who received PSZ–AFP at any time during aplasia or post-engraftment developed a b-IFD, compared to 6.3% (4/64) of patients under FCZ.
Roth RS [87] *Switzerland*	Monocenter, retrospective; *2010–2020;* 515 allo-HSCT recipients (28/515 already received a prior HSCT); Conditioning: MAC 28%, RIC 72%; Donor: 46% MUD, 31% MRD, 15% haplo, 8% MMUD.	AFP at HSCT: FCZ 47% (*n* = 244), mold-active azole 35% (*n* = 181 (123 PSZ, 55 VCZ, 3 ISA)), echinocandin 13% (*n* = 67), L-AMB 5% (*n* = 23); *Cumulative incidence of IMI after 1 year was **7%** (IA 5%, non-IA IMI 2%),* *48 (**9.3%**) patients developed 51 proven/probable IMI (34/51 IA, 9/51 mucormycosis, 8/51 other molds), 35/51 (69%) were b-IMIs (22 IA, 13 non-IA IMI) and AFP at b-IMIs diagnosis was mold active azoles (26/35), echinocandins (7/35), L-AMB (2/35)* Not available; *Ten proven IMI (four IA, five mucormycosis, one IMI due to other molds) and forty-one probable IMI (thirty IA, four mucormycosis, seven IMI due to other molds);* *Among IA: thirteen A. fumigatus, three A. terreus, three A. ustus, two A. niger;* *Mucormycosis: six Rhizomucor pusillus, two Rhizopus spp., one Absidia;* *Other molds: three Fusarium, one Alternaria, one Hormographiella aspergillata, one Scedosporium, one Schizophyllum commune, one Scopulariopsis.*	All-cause 1-year mortality was 33%: 48% and 31% in patients with and without IMI (*p* = 0.02); at 1 year post-IMI diagnosis, 71% of the patients were dead; *Prior allogeneic HSCT (s-HR 4.06), grade ≥2 a-GVHD (s-HR 3.52);* Overall, 29%, 21% and 50% of patients were diagnosed with IMI, respectively, during the first 30, 31–180, and >180 days post-HSCT; No azole-resistant *A. fumigatus* were identified; Mortality predictors included: disease relapse (HR 7.47), a-GVHD (HR 1.51), CMV serology–positive recipients (HR 1.47) and IMI (HR 3.94).
Bogler J [88] *USA*	Monocenter, retrospective; *2014–2018;* 305 allo-HSCT recipients; Conditioning: MAC 48%, RIC 45%; Donor: matched 67%, mismatched 26%, haplo 7%.	Propensity score matched cohort analysis: 210 HSCT recipients receiving VCZ–AFP, 95 HSCT recipients receiving ISA–AFP; *The incidence of proven/probable IFDs at day 180 was **2.9%** (6/210) and **3.2%** (3/95) in VCZ cohort and ISA cohort, respectively (p* = *0.88);* Not available; *In the VCZ cohort 5/6 IFDs occurred after AFP discontinuation (4 probable IA,1 C. glabrata candidemia) and 1/6 was a very early probable IA;* *In the ISA cohort all cases were b-IFDs: three candidemia (two C. parapsilosis, one C. glabrata).*	All-cause mortality at day 180 was 2.4% and 6.3% in the VCZ cohort and the ISA cohort, respectively (*p* = 0.09); two patients per cohort died because of IRM (one case in the ISA cohort was due to IFD (candidemia)); *Not available;* Duration of AFP was longer for ISA (median: 94 days (IQR: 87–100)) compared with VCZ (median: 76 days (IQR: 23–94)), while the time to engraftment and a-GVHD occurrence were superimposable; Premature AFP discontinuation was more frequent in the VCZ cohort (44%) vs. the ISA cohort (15%); the most common reason was biochemical hepatotoxicity (23% in VCZ, 5% in ISA); A trend towards more b-IFDs, particularly candidemia, was observed in the ISA cohort; however, the small number of events precluded formal comparisons.
Kraljevic M [89] *Switzerland*	Multicenter, retrospective; *2016–2018;* 288 allo-HSCT recipients; Conditioning: MAC 45%; Donor: MRD 28%, MUD 50%, MMUD 8%, haplo 13%.	Assessment of PSZ use (tablets, intravenous) in HSCT recipients: 194 (67%) PSZ–AFP, 94 (33%) PSZ–AFT, overall PSZ median duration of 90 days (IQR: 42–188.5); *The incidence rate of proven/probable b-IFDs was **3.1%** (9/288) without difference between PSZ AFP (7/194, 3.6%) and PSZ AFT (2/94, 2.1%);* There were 1944 PSZ measurements performed, with a median level of 1.3 mg/L (IQR: 0.8–1.96); the PSZ level was <0.7 mg/L in 20% when used for AFP and <1 mg/L in 31% when used for AFT; *Proven b-IFDs: two candidemia due to C. glabrata, two IMI (one Rhizomucor pusillus/Rhizopus, one A. ustus);* *Probable IFDs: one hepato-splenic candidiasis, four IMI (two Aspergillus spp., one Rhizomucor pusillus, one unidentified mold).*	Not assessed; *Not assessed;* There was no difference in the overall PSZ level mean values between patients with (1.32 mg/L) and without (1.47 mg/L) b-IFDs; No significant differences in liver function tests between baseline and end-of treatment; PSZ dose and formulation remained unchanged in 39% of patients; whereas 7% and 54% underwent formulation change only and dose ± formulation change, respectively; PSZ levels remain below target levels in 1/3 of patients: considering the low incidence of b-IFDs among patients with levels in the targeted range, routine PSZ TDM could be useful.
Stern A [90] *USA*	Monocenter, prospective; *2017–2018;* 95 allo-HSCT recipients (33% ex vivo T-cell depletion); Conditioning: MAC 56%, RIC 30%; Donor: MRD 20%, MUD 38%, mismatched related/unrelated 27%, haplo with PT-Cy 15%.	Open label single arm study: micafungin 150 mg daily from admission to day 7 post-HSCT, followed by ISA AFP through the maximum of 98 days post-HSCT; ***3.1%** of patients (3/95) developed b-IFD while on ISA AFP;* 110 ISA levels were obtained from 92 patients: the median ISA level was 3 μg/mL (IQR 2.2–4.2), no significant difference was found in ISA levels between patients with a grade ≥ 2 of gut a-GVHD (*n* = 27, median 2.7 μg/mL) and those without (*n* = 62, median 3.25 μg/mL); ISA level was available at IFI diagnosis in one patient and it was 1.4 μg/mL; *Three Proven IFDs: 3/3 cases of candidemia (two C. parapsilosis, one C. glabrata).*	Overall, 6.3% of patients died during the study (three during AFP phase, three during follow-up); The causes of death included: two candidemia (1/2 with concomitant persistent *E. faecium* bacteremia), two GVHD, one graft failure and one Pneumocystis jirovecii pneumonia; *Not assessed;* 45% of patients developed a-GVHD grade ≥ 2; 7.4% of patients stopped ISA AFP due to treatment emergent toxicity (mainly liver); *C. parapsilosis* strains were sensitive to FCZ (ISA sensitivity testing was not carried out).
Malagola M [91] *Italy*	Monocenter, retrospective; *2016–2021;* 200 allo-HSCT; Conditioning: MAC 54%, RIC 46%; Donor: MRD 28%, MUD 49%, haplo with PT-Cy 23%.	Cohort A (2016–2018) FCZ AFP; Cohort B (2019–2021) FCZ AFP in low-risk patients, PSZ AFP in high-risk patients plus aerosolized L-AMB (week 1: 2.5 mL/daily of a 5 mg/mL solution; week 2 until engraftment: 2.5 mL for 2 consecutive days every week); *The incidence of proven/probable IFDs was comparable: **13%*** vs. ***18%** in Cohort A and B, respectively;* Not reported; *By day 100 post-HSCT: four proven–probable IA, one Pityrosporum malassezia;* *From day 100 to day 180 post-HSCT: twenty proven–probable IA, three C. albicans, three P. jirovecii pneumonia.*	The OS was not affected by the development of IFDs in either cohort; *Not reported;* Even though patients in Cohort B were at higher risk of developing IFDs (more haplo transplants with more MAC), the extensive use of PSZ for AFP balanced this risk.
Busca A [92] *Italy*	Monocenter, retrospective; *2004–2020;* 563 allo-HSCT recipients; Conditioning: MAC 69%, RIC 31%; Donor: MRD 34%, MUD with ATG 51%, haplo with PT-Cy 15%.	AFP: FCZ in MRD and MUD (*n* = 441), micafungin in haplo (*n* = 62), MAP *n* = 9, secondary AFP *n* = 42; *At 30, 180, and 365 days post-HSCT, the cumulative incidences of proven/probable IFDs were **4.1%**, **8.1%**, and **9.6%**, respectively (58 cases (n* = *47 probable; n* = *11 proven));* *At 1 year, the cumulative incidence of IFDs was 3.2% for MRD, 11.4% for MUD and 16.8% for haplo;* Not assessed; *Molds were the predominant agents (n* = *fifty Aspergillus; n* = *one Mucor), followed by IC (n* = *five non-albicans Candida; n* = *one C. albicans; n* = *one Trichosporon).*	Overall survival at 1 year was 33% vs. 55% in patients with IFDs compared to those without (*p* < 0.001); overall, the IFD-related mortality rate was 21%, while it was 17% and 36% in patients with probable and proven IFDs, respectively; *Donor type (haplo* vs. *MUD vs. MRD) was a predictive factor for IFD (s-HR 1.91, 95% IC 1.13–3.20);* Only 35% of patients with IFDs were neutropenic at the time of infection; L-AMB was used in the majority of patients receiving diagnostic-driven therapy (38%), while caspofungin was administered more frequently in those receiving fever-driven strategy (43%).
Lindsay J [93] *USA*	Monocenter, retrospective; *2018–2020;* 381 allo-HSCT, 276 ASCT, 153 CAR-T recipients; Conditioning and donor not reported.	Assessment of VCZ use in 6 months before or after cellular therapies: 102/381 HSCT (40/102 in the first 30 days after HSCT), 13/276 ASCT and 10/153 CAR-T recipients; ***Not assessed;*** VCZ TDM levels: 52% of patients had a therapeutic level (1.0–5.5 mg/L) at the first measurement (median 2.8 mg/L (range 0.1–13.5)) at a median of 6 days of therapy, 26% sub-therapeutic, 21% supra-therapeutic; 69% of patients had a therapeutic level at the third measurement (2.3 mg/L (0.1–7.7)) at a median of 29 days; *Not assessed.*	Not assessed; *Not assessed;* Predictive factors associated with sub- or supra-therapeutic levels at multivariable analysis: among BMI ≥ 30, concurrent omeprazole, concurrent letermovir, indication for VCZ (AFP vs. AFT) and history/timeframe of HSCT; the only significant association was lower odds of a supra-therapeutic VCZ level among those undergoing HSCT within the previous 30 days, compared to those without a history of HSCT. Therapeutic level attainment of VCZ remains a challenge.
Lerolle N [94] *France*	Monocenter, retrospective; *2007–2010;* 270 patients: 168 AML receiving induction chemotherapy, 96 allo-HSCT recipients with a-GVHD and 4 idiopathic aplastic anemia; Conditioning and donor not reported.	All patients received PSZ AFP oral suspension; ***3.2%** of patients (9/270) developed IFDs (seven AML, one allo-HSCT recipients, one aplastic anemia);* Sub-optimal exposure to PSZ oral suspension was frequent (5/7 patients with IFDs had the first PPC < 0.5 mg/L (3/5 the first PPC < 0.3 mg/L)); *Proven IFDs (6/9): two candidemia (two C. glabrata), two disseminated fusariosis (two F. solani), one pulmonary mucormycosis, one IA (A. terreus);* *Probable IFDs (3/9): one pulmonary mucormycosis, two IA.*	Outcome of patients with b-IFDs: five were cured, one died due to AML, three died due to IFD (two IA, one fusariosis); Overall, 33% of deaths (3/7) were IFD-related; *Compared with patients who did not experience IFD, the single significant risk factor was a first PPC < 0.3 mg/L (ref ≥0.5 mg/L, RR 7.90 (1.32–47.3));* Amongst 6/9 patients with positive mycological cultures, 5 were tested for susceptibility to antifungals and 4/5 isolates had high MIC to PSZ (including both Candida); L-AMB was the most common first-line AFT (8/9 patients); Median duration of PSZ AFP at IFD onset was 18 days (range: 7–126).
Fontana L [52] *USA*	Monocenter, retrospective; *2016–2018;* 145 AML patients (63/145 allo-HSCT recipients); Conditioning: MAC 69%, RIC 31%; Donor: MRD 34%, MUD with ATG 51%, haplo with PT-Cy 15%.	All patients received ISA as primary AFP; ***7.6%** of patients (11/145) developed proven/probable b-IFDs, 1/11 occurred post-HSCT (probable IA);* Not routinely assessed, ISA trough levels were obtained within 72 h of b-IFD in 5/11 cases (5/5 breakthrough IA) with a median of 3.7 μg/mL (range, 3.3–6.3 μg/mL); *Probable b-IFDs: six probable IA, one probable mucormycosis (Syncephalastrum racemosum), one probable Fusarium dimerum infection;* *Proven b-IFDs: two fungemia (one C. glabrata, one Fusarium spp.), one proven mucormycosis (Rhizopus microspores).*	At 42 days after b-IFDs diagnosis, 55% of patients (6/11) had died; *Not assessed;* No b-IFD occurred among HSCT recipients receiving high-dose steroids while receiving AFP; ISA phenotypic antifungal susceptibility testing could be performed on one *A. fumigatus* isolate (MIC = 0.5 μg/mL) and one *S. monosporum/racemosum* isolate (MIC = 2 μg/mL); CYP51A gene sequencing performed on *A. fumigatus* DNA found in three bronco-alveolar lavage fluid samples did not demonstrate mutations associated with azole resistance; In induction and re-induction chemotherapy the occurrence of b-IFDs was analyzed according to the type of MAP: b-IFDs complicated 10.2% of ISA, 4.1% of PSZ, and 1.1% of VCZ courses among patients with de novo or relapsed/refractory AML, with IA complicating 6.8% of ISA, 1.3% of PSZ, and 0% VCZ courses.
Posteraro B [2] *Italy*	Multicenter, retrospective; *2011–2015;* 133 patients with hematological malignancy and candidemia; Allo-HSCT recipients 17%.	Overall, 42% of patients (56/133) experienced a breakthrough candidemia, receiving AFP (*n* = twenty-four FCZ, *n* = twenty PSZ, *n* = five ITZ, *n* = four echinocandin, *n* = two L-AMB, *n* = one VCZ); ***Not applicable;*** Not available; *Among breakthrough candidemia, the most common species were C. parapsilosis (19/56, 3/19 echinocandin AFP) and C. krusei (10/56, 10/10 azoles AFP);* *Non-albicans Candida species were the mostly isolated species (67% (n* = *35 C. parapsilosis, n* = *16 C. glabrata, n* = *14 C. krusei, n* = *13 C. tropicalis, n* = *11 uncommon species)) and C. albicans caused the remaining 33% of episodes;* *FCZ resistance was present in 25/133 (fourteen C. krusei, six C. glabrata, two C. parapsilosis, two C. albicans, one C. tropicalis).*	The 30-day mortality was 36.1%, with 25/48 deaths being classified as *Candida*-related; *Not available;* All patients with breakthrough candidemia tended to develop *C. krusei* infection (10/56, *p* = 0.02) or an FCZ-resistant infection (14/50, *p* = 0.04), compared to patients who did not have breakthrough candidemia (4/77 and 10/77, respectively).
Puerta-Alcalde P [95] *Spain*	Multicenter, prospective; *2017–2020;* Adults with hematological malignancy prospectively enrolled; Among 121 b-IFDs (41 proven, 53 probable, 27 possible): 55% AML, 69% grade IV neutropenia, 42% allo-HSCT, 20% ICU in previous 30 days.	Antifungal drug at the diagnosis of b-IFD (10/121 b-IFD while on combination of antifungals): 32% PSZ (tablets or intravenous), 29% echinocandins, 25% FCZ, 10% Amphotericin B, 5% ISA, 5% VCZ; ***Not applicable;*** Not systematically reported; however, among microbiologically documented b-IFDs, any fungus susceptible to the prior antifungal administered at good therapeutic levels was isolated (except for some catheter-related fungemia); *Proven b-IFDs: forty-one fungi recovered from blood (n* = *twenty-three Candida (six C. krusei, five C. parapsilosis, four C. glabrata, three C. albicans, two C. guilliermondii, one C. tropicalis, one C. orthopsilosis, one C. kefyr), n* = *two Geotrichum, n* = *one Trichosporon asahii, n* = *one Rhodotorula mucilaginosa, n* = *one Magnusiomyces capitatus),* eight *positive culture of a sterile site (n* = *two F. solani, n* = *one Rhizopus, n* = *one A. flavus, n* = *one Cunninghamella, n* = *one A. niger, n* = *one C. krusei, n* = *one C. guilliermondii), seven histopathological findings of a sterile specimen (n* = *four Mucorales, n* = *one A. flavus, n* = *one A. fumigatus, n* = *one unidentified mold);* *Probable b-IFDs: forty-eight IA, two scedosporiosis, two paecilomycosis, one mixed A. niger and P. lilacinum.*	100-day mortality was 47% (b-IFD was the primary or essential cause of death in 61%), and a higher mortality trend was observed in those episodes receiving inappropriate empiric AFT (66.7% vs. 44.3%); The highest mortality was seen in mucormycosis and other rare yeasts; *Not applicable;* At b-IFD diagnosis, antifungal therapy was mostly changed (91%), mainly to L-AMB (49%); Molecular testing of biopsy samples identified a high number of non-*Aspergillus* molds; Causes of b-IFDs: *n* = 63 poor activity of prior antifungal (*n* = 27 intrinsically resistant, *n* = 9 antifungal non-susceptibility, *n* = 12 acquired resistance, *n* = 15 b-IA on echinocandins (despite no in vitro resistance)), *n* = 7 presence of factor perpetuating the fungemia (intravenous catheter), *n* = 4 inappropriate dosages (3 L-AMB 1 mg/kg/day, 1 sub-therapeutic VCZ level); Among 55 proven/probable IA: 11/55 *A. fumigatus* (0/11 azole-resistant); 18/55 non-fumigatus *Aspergillus* (seven *A. terreus*, four *A. flavus*, four *A. niger*, one *A. ustus*, one *A. alliaceus*, one *A. hiratsukae*); 26/55 diagnosis with galactomannan.
Hong Nguyen M [96] *USA*	Multicenter, prospective; *2017–2020;* 1177 immuno-compromised adults receiving MAP with triazoles; Hematological malignancy 76.5%, allo-HSCT 33%, ICU 20%, SOT 15%.	22% ISA (*n* = 256), 34% PSZ (*n* = 397), 23% VCZ (*n* = 272), 21% multiple/sequenced MAP (*n* = 252); ***4.9%** (51/1030) probable/proven b-IFDs (**4.1%** (9/221) ISA, **4.0%** (15/374) PSZ, **1.3%** (3/226) VCZ, **11.5%** (24/209) multiple/sequenced MAP);* TDM was measured in 34.6% (407/1176) of patients (7.8% (20/256) ISA, 37.9% (150/397) PSZ, 43.4% (118/ 272) VCZ) and median TDM levels were: 3.5 mg/mL (0.4–9.4) for ISA, 1.4 mg/mL (0.1–6.5) for PSZ, 2.0 mg/mL (0.1–17.8) for VCZ; *35 proven b-IFD, 16 probable b-IFD (type of proven/probable b-IFD not clearly classified (authors reported that the majority of b-IFDs were sustained by Candida and Aspergillus)).*	Among patients with b-IFDs (including possible b-IFD), all-cause mortality and fungal-specific mortality at the end of MAP were 11% (8/73) and 4.1% (3/73); *Not reported;* Discontinuation of MAP due to adverse drug reaction was reported in 11.1% of patients (2.0% (5/245) ISA, 8.2% (30/368) PSZ, 10.1% (27/267) VCZ).
Scott SA [97] *USA*	Monocenter, retrospective; *2015–2021;* 126 adults with hematological malignancy (96 allo-HSCT recipients, 30 induction chemotherapy for AML); 61% of patients (77/126) with a-GVHD.	Matched-control study 1:2, 42 AFP with ISA, 84 AFP with VCZ (3) or PSZ (81), patients receiving secondary AFP were significantly more represented in ISA cohort (69% versus 26%); *The incidence of proven/probable b-IFDs was **7%** (3/42) in the ISA group compared to **6%** (5/84) in the PSZ–VCZ group;* Not available; *Three proven IFD, five probable IFD (type of b-IFDs not precisely reported (in the ISA group there was one Zygomycetes and one Aspergillus spp., in the PSZ–VCZ group there was one Candida spp. and three Aspergillus spp.)).*	One patient in each group died of b-IFD; *Not reported;* Patients in the PSZ–VCZ group experienced more hepatotoxicity, compared to patients receiving ISA (16.7% vs. 4.8%).

Abbreviations: allo-HSCT, allogeneic hematopoietic stem cell transplant; MAC, myeloablative conditioning; RIC, reduced intensity conditioning; AFP, antifungal prophylaxis; PSZ, posaconazole; VCZ, voriconazole; FCZ, fluconazole; IFD, invasive fungal disease; IA, invasive aspergillosis; b-IFD, breakthrough invasive fungal disease; b-IA, breakthrough invasive aspergillosis; IC, invasive candidiasis; ICU, intensive care unit; a-GVHD, acute graft-versus-host disease; TAM, transplant-associated micro-angiopathy; MUD, matched unrelated donor; haplo, haploidentical donor; MMUD, mismatched unrelated donor; MRD, matched related donor; L-AMB, liposomal amphotericin B; IMI, invasive mold infections; b-IMI, breakthrough invasive mold infections; s-HR, sub hazard ratio; HR, hazard ratio; ISA, isavuconazole; IRM, infection-related mortality; IQR, interquartile range; AFT, antifungal therapy; PT-Cy, post-transplant cyclophosphamide; OS, overall survival; ATG, anti-thymocyte globulin; MAP, mold-active prophylaxis; ASCT, autologous stem cell transplantation; CAR-T, chimeric antigen receptor T-cell; TDM, therapeutic drug monitoring; BMI, body mass index; AML, acute myeloid leukemia; PPCs, plasmatic posaconazole concentrations; SOT, solid organ transplant.

**Table 3 microorganisms-12-00117-t003:** Characteristics of chemotherapy treatments, posaconazole plasma concentration data, follow-up time and causes of death.

		Overall (*n* = 339)	Induction (*n* = 153)	Consolidation (*n* = 126)	Salvage (*n* = 60)	*p* Value
Patients’ age at CT start, years, median [IQR]		61.5 (44.6; 69.5)	62.6 (46.4; 70.0)	62.6 (43.8; 69.9)	54.0 (41.1; 67.1)	0.188
Number of total treatment cycles		1052	457	424	171	-
MAP	PSZ	339 (100%)	153 (100%)	126 (100%)	126 (100%)	-
ANC ≤500 for ≥7 days before CT start	Yes	86 (25.7%)	54 (36.2%)	9 (7.14%)	23 (38.3%)	**<0.001**
CT type	3 + 7 or Vyxeos	69 (22.9%)	69 (54.3%)	-	-	-
	Fludarabine-based	46 (15.3%)	46 (36.2%)	-	-	
	Idarubicin–Cytarabine–Etoposide	12 (3.99%)	12 (9.45%)	-	-	
	High dose Ara-C	87 (28.9%)	-	87 (76.3%)	-	
	Intermediate dose Ara-C	27 (8.97%)	-	27 (23.7%)	-	
	Salvage chemotherapy	43 (14.3%)	-	-	43 (71.7%)	
HMAs	No	292 (86.1%)	127 (83.0%)	114 (90.5%)	51 (85.0%)	0.018
	Yes	28 (8.26%)	14 (9.15%)	11 (8.73%)	3 (5.00%)	
	Yes, combined with VEN	19 (5.60%)	12 (7.84%)	1 (0.79%)	6 (10.0%)	
Target therapy	No	310 (91.4%)	143 (93.5%)	120 (95.2%)	47 (78.3%)	**<0.001**
	Yes, combined with CT	19 (5.60%)	8 (5.23%)	6 (4.76%)	5 (8.33%)	
	Yes, combined with HMAs	2 (0.59%)	2 (1.31%)	0 (0.00%)	0 (0.00%)	
	Yes, alone	8 (2.36%)	0 (0.00%)	0 (0.00%)	8 (13.3%)	
Type of target therapy	Dasatinib	1 (5.00%)	1 (10.0%)	0 (0.00%)	0 (0.00%)	0.069
	Gemtuzumab	9 (45.0%)	4 (40.0%)	5 (83.3%)	0 (0.00%)	
	FLT3 target therapy	10 (50.0%)	5 (50.0%)	1 (16.7%)	4 (100%)	
ANC engraftment [11,40]	Yes	290 (86.3%)	131 (87.3%)	119 (94.4%)	40 (66.7%)	**<0.001**
Reinduction CT	Yes	-	35 (23.5%)	-	-	-
Type of reinduction CT	Fludarabine-based	-	31 (88.6%)	-	-	-
	Others	-	4 (11.4%)	-	-	
ANC engraftment post reinduction CT	Yes	-	33 (94.3%)	-	-	-
G-CSF use	Yes	170 (50.1%)	58 (37.9%)	93 (73.8%)	19 (31.7%)	**<0.001**
RBC transfusion, median (IQR)		4.00 [0.00; 10.5]	9.00 [2.00; 13.0]	3.00 [0.00; 5.00]	2.00 [0.00; 8.25]	**<0.001**
Platelets transfusion, median (IQR)		3.00 [0.00; 8.00]	5.00 [0.00; 10.0]	2.00 [0.00; 4.00]	4.00 [0.00; 8.25]	**<0.001**
Hematological response to CT	CR	248 (78.7%)	111 (78.7%)	106 (89.1%)	31 (56.4%)	**<0.001**
	NR	67 (21.3%)	30 (21.3%)	13 (10.9%)	24 (43.6%)	
b-IFD	Probable b-IFD	6 (1.77%)	2 (1.31%)	1 (0.79%)	3 (5.00%)	0.155
	Proven b-IFD	0 (0.00%)	0 (0.00%)	0 (0.00%)	0 (0.00%)	
At least one PPC	Yes	323 (95.3%)	148 (96.7%)	117 (92.9%)	58 (96.7%)	0.319
Number of consecutive PPCs < 0.7 mg/L per treatment [7,8]	0	246 (72.6%)	106 (69.3%)	98 (77.8%)	42 (70%)	0.424
	1	68 (20.1%)	32 (20.9%)	24 (19.1%)	12 (20%)	
	≥2	22 (6.5%)	12 (7.84%)	4 (3.18%)	6 (10%)	
Time from leukaemia diagnosis to CT start, months, median [IQR]		1.45 [0.40; 2.76]	0.39 [0.20; 0.76]	2.22 [1.64; 3.05]	4.51 [0.92; 10.6]	**<0.001**
Time from CT start to b-IFD, months, median [IQR]		0.87 [0.49; 1.57]	3.40 [2.57; 4.24]	0.43 [0.43; 0.43]	0.69 [0.44; 0.87]	0.172
Time from CT start to next CT treatment, death or end of follow-up, months, median [IQR]		1.91 [1.41; 3.67]	1.84 [1.28; 3.95]	1.92 [1.48;3.48]	2.09 [1.61; 3.42]	0.242
Death	Yes	35 (19.0%)	11 (7.3%)	8 (8.9%)	16 (29.6%)	**<0.001**
Causes of death	Disease	24 (68.6%)	8 (66.7%)	4 (50%)	12 (75%)	0.534
	CT-related toxicities	2 (5.7%)	1 (8.3%)	0 (0.0%)	1 (6.3%)	
	Infection-related death	9 (25.7%)	2 (16.7%)	4 (50%)	3 (18.7%)	
	Other	1 (2.9%)	1 (8.3%)	0 (0.0%)	0 (0.0%)	

Abbreviations: ANC, absolute neutrophil count; Ara-C, cytarabine; b-IFD, breakthrough invasive fungal disease; CR, complete remission; CT, chemotherapy; G-CSF, granulocyte colony-stimulating factor; HMA, hypomethylating agent; IQR, interquartile range; NR, non-response; PPC, posaconazole plasma concentrations; PSZ, posaconazole; RBC, red blood cells; VEN, venetoclax. Results described by median (first–third quartile) or frequency (%); *p*-values by chi-square or Fishers’ exact test and Wilcoxon rank-sum test, as appropriate.

**Table 4 microorganisms-12-00117-t004:** Characteristics of patients diagnosed with probable pulmonary invasive aspergillosis *.

	Age, Years	Sex	High-Risk Neutropenia before CT	Chemotherapy Treatment	Chemotherapy Type	Response to CT	Time from CT to b-IFDs, Days	PPCs at b-IFD Diagnosis	Posaconazole Formulation	Antifungal Therapy	b-IFD Outcome at 6 Weeks	b-IFD Outcome at 12 Weeks	Death, Cause	Outcome after b-IFD
1	74	Male	No	Induction, re-induction	3 + 7, Flag-Ida	CR	53	Inadequate (PPC < 0.7 mg/L)	Suspension	Voriconazole	Persistent IFD with IR	Treatment success	No	He underwent consolidation chemotherapy
2	23	Male	Yes	Salvage	Flag-Ida	CR	6	Not available	Not known	Voriconazole	Persistent IFD with IR	Treatment success	Yes, disease	He underwent allo-HSCT; then, relapse
3	63	Male	Yes	Salvage	VEN + HMA	NR	32	Adequate (PPC > 1 mg/L)	Tablets	L-AMB	Persistent IFD without IR	Treatment success	No	He continued VEN + HMA, then underwent allo-HSCT
4	72	Male	No	Consolidation	Intermediate dose Ara-C	CR	13	Adequate (0.7 < PPC < 1 mg/L)	Suspension	Voriconazole	Treatment success	Treatment success	No	He continued consolidation, then underwent allo-HSCT
5	76	Male	No	Salvage	Flag-Ida	CR	21	Adequate (PPC > 1 mg/L)	Tablets	Voriconazole	Treatment success	Treatment success	Yes, disease	He underwent consolidation; then, relapse
6	66	Male	No	Induction	VEN + HMA	CR	154	Inadequate (PPC < 0.7 mg/L)	Tablets	Voriconazole	Persistent IFD without IR	Treatment success	No	He continued VEN + HMA

Abbreviations: CT, chemotherapy; VEN, venetoclax; HMA, hypomethylating agents; Ara-C, cytarabine; CR, complete response; NR, non-response; b-IFD, breakthrough invasive fungal disease; L-AMB, liposomal amphotericin B; IR, immune-reconstitution; allo-HSCT, allogeneic hematopoietic stem cell transplantation. * Mycological criteria: 6/6 galactomannan positivity. No positive cultures.

## Data Availability

Additional data related to this paper are available and may be requested to the corresponding author.

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
