# Peer review of "High-Risk Neutropenic Fever and Invasive Fungal Diseases in Patients with Hematological Malignancies"

_microorganisms, 2024, doi:10.3390/microorganisms12010117_

Round 1

Reviewer 1 Report

Comments and Suggestions for Authors

Thank you for submitting your manuscript titled "High-Risk Neutropenic Fever and Invasive Fungal Diseases in Patients With Hematological Malignancies". I have carefully reviewed the content, and overall, the paper discusses the prevention, diagnosis, and treatment of invasive fungal diseases (IFD) in patients with hematologic malignancies. The article is comprehensive, presenting a retrospective single-center experience. However, there are some issues that need to be addressed. Therefore, major revisions are required before considering acceptance for publication.

Major Comments:

Upon overall evaluation, the length of the manuscript is excessive, with unnecessary redundancies and repetitive descriptions. I recommend streamlining the content for better readability and suggest language polishing to enhance overall clarity.

Minor Comments:

For Subheading 1: "Diagnosis of invasive fungal diseases, definition of breakthrough invasive fungal diseases and advances in microbiological diagnostic tools," the scope may be too broad, resulting in overly complex content. I recommend splitting this subheading to improve readability. Additionally, the author extensively describes published research from line 69 to 153 and lines 159 to 220, as well as 242 to 274. Please consider condensing the language for more concise and accurate descriptions.

Similarly, the detailed descriptions of others' research from lines 159 to 226 seem excessive. Emphasize your innovative content, as seen in the lines following "Thus, consensus definitions of persistent, refractory, relapsed, and b-IFDs..." Highlight your contributions rather than providing detailed descriptions of existing literature.

The use of tables to present "further studies on IFDs and AFP practice in allo-HSCT recipients were published" is effective; therefore, avoid unnecessary repetition in the main text. Consider streamlining the description from lines 405 to 481.

In the "Single-center experience with posaconazole prophylaxis in patients with acute myeloid leukemia" section, the descriptions under Methods, Study Objectives, and Statistical Analysis need more clarity. Ensure that definitions for each threshold are supported by appropriate references. In Table 3, include references for the Ref index to enhance the readability of the results presentation.

Comments on the Quality of English Language

Overall, the length of the article is excessive, with some unnecessary redundancies and repetitive descriptions. There are numerous complex sentences containing professional terminology. It is recommended to refine the language to enhance the readability of the article.

Reviewer 2 Report

Comments and Suggestions for Authors

This is a systematic review considering advances on prophylaxis, diagnosis and treatment of invasive fungal diseases in patients with neutropenic fever and haematological malignancies (HM).

It is an important topic once IFDs remain an important cause of mortality. Results from the review are well exposed, in details.

If authors need to consider the size of the manuscript in order to shorten it, some parts of results that are also in the table, can be excluded from the text.

Reviewer 3 Report

Comments and Suggestions for Authors

a systematic review [however this reads more like a narrative review] with single-center retrospective experience [which starts at line 482 – although this needs a clear heading of ‘our single-center experience’.

The abstract needs specific data relating to the single-center experience – number of patients etc.

Overall the text is poorly written and poorly structured and possibly too long and confusing.

Line 53-66 needs to be titled “Search strategy”

Line 67 to 81 is confusing and appears out of place given the heading “Diagnosis of invasive fungal diseases, definition of breakthrough…”

“Expansion of …” at each appearance is unclear, do you mean “Key…”

Table 4  had not loaded properly in my copy

Comments on the Quality of English Language

A better layout with use of clear headings

Round 2

Reviewer 1 Report

Comments and Suggestions for Authors

The author responded diligently and made revisions to the questions raised.

Comments on the Quality of English Language

There has been an improvement in English language proficiency.

Reviewer 3 Report

Comments and Suggestions for Authors

Many thanks for rendering the manuscript more concise